# The Splicing Factor OsSCL26 Regulates Phosphorus Homeostasis in Rice

**DOI:** 10.3390/plants12122326

**Published:** 2023-06-15

**Authors:** Shanshan Lu, Jun Ye, Hui Li, Fengyu He, Yue Qi, Ting Wang, Wujian Wang, Luqing Zheng

**Affiliations:** College of Life Sciences, Nanjing Agricultural University, Nanjing 210095, Chinat2021018@njau.edu.cn (W.W.)

**Keywords:** rice, splicing factor, *osscl26* mutant, phosphorus homeostasis

## Abstract

Phosphorus (P) is an essential nutrient for plant growth. However, its deficiency poses a significant challenge for crop production. To overcome the low P availability, plants have developed various strategies to regulate their P uptake and usage. In this study, we identified a splicing factor, OsSCL26, belonging to the Serine/arginine-rich (SR) proteins, that plays a crucial role in regulating P homeostasis in rice. *OsSCL26* is expressed in the roots, leaves, and base nodes, with higher expression levels observed in the leaf blades during the vegetative growth stage. The OsSCL26 protein is localized in the nucleus. Mutation of *OsSCL26* resulted in the accumulation of P in the shoot compared to the wild-type, and the dwarf phenotype of the *osscl26* mutant was alleviated under low P conditions. Further analysis revealed that the accumulated P concentrations in the *osscl26* mutant were higher in the old leaves and lower in the new leaves. Furthermore, the P-related genes, including the PHT and SPX family genes, were upregulated in the *osscl26* mutant, and the exclusion/inclusion ratio of the two genes, *OsSPX-MFS2* and *OsNLA2*, was increased compared to wild-type rice. These findings suggest that the splicing factor OsSCL26 plays a pivotal role in maintaining P homeostasis in rice by influencing the absorption and distribution of P through the regulation of the transcription and splicing of the P transport genes.

## 1. Introduction

Phosphorus (P) is an essential macro-nutrient for plants, serving as a key component of nucleic acids, lipids, ATP, ADP, and sugars. It also participates in important physiological processes such as photosynthesis, electron transfers, oxidative phosphorylation, organic metabolism, and enzymatic reactions [1]. As a result, an adequate P content plays a crucial role in plant growth and crop yields [2]. However, in soil, P can become adsorbed, precipitated, or transformed into organic forms, making it difficult for plants to absorb and utilize it effectively, thus affecting normal growth and reproduction. To address this, many farmers use phosphate fertilizers to meet crop P demands. However, the primary source of phosphate fertilizer is phosphate rock, a non-renewable resource that hinders sustainable agricultural development [3]. Rice is one of the most important crops worldwide, feeding nearly half of the world’s population. Understanding the regulation mechanisms of P homeostasis in rice can not only help cultivate rice varieties with a high P efficiency to increase food production but also provide a theoretical basis for building environmentally friendly and sustainable agriculture.

Rice possesses adaptive mechanisms to cope with low P environments, primarily consisting of two components. Firstly, in response to a P deficiency, the root system of rice undergoes changes, such as an increased root length and surface area, accompanied by a reduction in the root diameter, thereby enabling better phosphate absorption from the soil [4]. Rice also induces the secretion of acid phosphate and root-specific exudates while activating high-affinity phosphate transporters. The excretion of malate and citrate by the roots facilitates the dissociation of inorganic phosphorus (Pi) from soil minerals, thereby enhancing the P uptake efficiency [5]. The upregulation of high-affinity P transporter genes in rice, such as *OsPT1,* facilitates P absorption [6,7]. Additionally, under low P stress, rice roots can establish mutualistic symbiotic associations with fungi in the soil, enhancing the contact surface between the roots and the soil and facilitating the cooperative uptake of inorganic P nutrients [8]. Secondly, rice redistributes the absorbed P from older leaves to newly developing leaves, promoting new leaf growth and enhancing the P utilization efficiency. The plant vacuoles serve as the primary storage sites for P. When faced with P stress, the P stored in the vacuoles is transported to other organs. The SPX-MFS family of proteins in rice is involved in vacuolar P transport [9,10]. Under an extreme P deficiency, *OsPHT1;3* exhibits high expression levels and facilitates P uptake while translocating Pi from the roots to the shoots. Notably, it is specifically expressed in the phloem of the basal vascular tissue that is responsible for the Pi transfer from the older leaves to the younger ones [11]. OsPHT1;8, on the other hand, acts as a high-affinity phosphate transporter, facilitating Pi transport from the vegetative organs to the reproductive organs during the later stages of plant growth [12]. OsPHO1;2 mediates the Pi transport between the root and stem tissues of rice during the seedling stage [13]. The efficient transport of Pi from the vegetative organs to the reproductive organs significantly improves the P recycling efficiency, promoting P accumulation in the later stages of crop growth and development, and ultimately enhancing the crop seed setting rates [13].

In all eukaryotes, pre-mRNA undergoes splicing, a process that removes introns and joins exons to form mature mRNA. This process is primarily executed by the splicing complex, with alternative splicing (AS) occurring when the complex splicing site recognition is altered [14,15]. Alternative splicing not only regulates plant growth, development, and stress responses but also plays a crucial role in plant nutrient absorption and utilization. Numerous genes in Arabidopsis, rice, barley, and maize have been identified to undergo alternative splicing events [16,17,18]. For instance, the enhanced translation of Z1F2, resulting from alternative splicing in Arabidopsis, leads to an increased resistance to zinc stress [19]. An iron deficiency induces various splicing events in the *Arabidopsis thaliana* roots, where multiple genes are involved in the biological processes related to the P deficiency response [20]. Previous studies have identified several SR proteins as important regulators of zinc, manganese, and P in our laboratory, and they play a key role in regulating the nutrient balance of mineral elements in rice [21].

Serine/arginine-rich (SR) proteins are one of the major splicing factor families participating in various biological processes, including plant development and stress responses [22]. The deletion of the *SC35* and *SCL* subfamily results in shortened roots, serrated leaves, and other features, and also affects the *FLC*, an important gene for flowering [23,24]. Previous studies have found that mutations in *osscl25* and *osscl57* lead to the accumulation of P in the shoots. The specific function of the OsSCL26 protein of the same SCL subfamily in regulating the nutrient elements is still unclear, and whether it is involved in P homeostasis in rice is unknown. In this study, through analyzing the protein structure, tissue, and subcellular localization of OsSCL26, as well as its physiological responses to the mutant in P-deficient and normal conditions, we conclude that OsSCL26 plays an important role in maintaining P homeostasis in rice.

## 2. Results

### 2.1. Bioinformatics Analysis of OsSCL26 

*OsSCL26* is located on rice chromosome 3, encoding a protein with 219 amino acids. The ProtParam analysis showed that the predicted molecular weight of the OsSCL26 protein was 25.7 kDa. The NetPhos analysis revealed that OsSCL26 had 40 phosphorylation modification sites, including 33 serine phosphorylation sites, one threonine phosphorylation site, and six tyrosine phosphorylation sites. The ProtScale analysis showed that the OsSCL26 protein had the highest score of 1.267 at position 106 and the lowest score of −3.189 at position 118. Meanwhile, the number of amino acids with scores greater than 0 was significantly lower than the number of amino acids with scores less than 0. The area of the hydrophobic region was smaller than that of the hydrophilic region, indicating that the OsSCL26 protein was a hydrophilic protein (Figure 1A). The TMHMM Serverv.2.0 analysis showed that the OsSCL26 had no transmembrane structural domain, with no clear signal peptide (Figure 1B). The conservative structural domain analysis showed that the OsSCL26 protein contained an RNA recognition motif (RRM) structural domain (Figure 1C,D). The secondary structure predictions showed that the OsSCL26 protein was mainly composed of an alpha helix (19.27%), a beta fold (4.59%), an irregularly coiled strand (67.89%), and an extended strand (8.26%) (Figure 1E). The tertiary structure build mode was 2leb.1A, the model coverage was 39%, and the sequence similarity was 42% (Figure 1F).

### 2.2. Expression Patterns and Subcellular Localization of OsSCL26 

At the reproductive growth stage, *OsSCL26* was predominantly expressed in the leaves, including both the leaf blades and the leaf sheaths. The expression level of *OsSCL26* was higher in the above-ground organs than in the roots at various growth stages. The highest expression level of *OsSCL26* was observed in the old leaves and old leaf sheaths during the flowering stage (Figure 2A).

To investigate the detailed expression pattern of *OsSCL26*, we generated *Pro_OsSCL26_*-1300-GUS transgenic lines. After GUS staining, the results demonstrated that *OsSCL26* was expressed in the root crown, elongation zone, and maturation zone of the primary roots as well as in the lateral roots (Figure 2B(b1,b2)). The cross sections of the primary root showed that *OsSCL26* was expressed in the cortex(Figure 2B(b3,b4)). *OsSCL26* was also expressed in the stem and leaves of the shoots(Figure 2B(b5,b6,b9,b10)), and the cross sections of the stem and leaves indicated that *OsSCL26* was expressed in the vascular bundles(Figure 2B(b11,b12)).

To determine the subcellular localization of OsSCL26 in rice, we expressed OsSCL26-GFP in tobacco epidermal cells and rice protoplasts. The results showed that the GFP signal of OsSCL26-GFP was exclusively localized in the nucleus, while the GFP signal of the control vector 35S::GFP was expressed in both the nucleus and cytosol (Figure 2C,D). These results suggest that OsSCL26 is localized in the nucleus of rice. 

### 2.3. Phenotypic Analysis of the osscl26 Mutant at Different P Conditions

To investigate the role of *OsSCL26* in the regulation of P homeostasis in rice, we generated an *osscl26* mutant using CRISPR-Cas9 gene editing. The gRNA targeted the 2nd exon of *OsSCL26* and caused a C deletion, which resulted in a code shift mutation and affected the normal protein translation (Figure 3A–C). The wild-type and *osscl26* mutant were cultivated in nutrient solutions containing LP (low P: 10 μM Pi), MP (sufficient P: 91 μM Pi), and HP (high P: 450 μM Pi) for approximately 14 days. The shoot heights and dry weights of the *osscl26* mutants were significantly lower compared to the wild-type in all three P conditions (Figure 3D–F). Similarly, under both the LP and MP conditions, the root lengths and dry weights of the *osscl26* mutants decreased compared to the wild-type (Figure 3D–F). We then measured the inorganic P (Pi) and total P concentrations of the *osscl26* mutant, and the results showed that both the Pi and total P were significantly higher in the shoots of the mutant than that of the wild-type (Figure 3G,H).

### 2.4. Phosphate Uptake and Remobilization in osscl26 Mutants

To further determine the role of OsSCL26 in regulating the P distribution in leaves under different time periods and P levels, we performed a time-dependent P mobilization assay. We monitored the Pi concentration in each leaf and the stem at day 10, 16, 22, and 28. We found that the Pi concentration of the *osscl26* mutant had no significant difference compared to the wild-type in leaf2 and the stem at 10 d (Figure 4A). Under the LP and MP conditions, the Pi concentration of *osscl26* in leaf2 and stem were higher than the wild-type at 16 d. Under the HP conditions, the Pi concentration of *osscl26* in leaf2, leaf3, and stem were higher than the wild-type at 16 d (Figure 4B–D). Under different P supply conditions, there was a gradual decrease in the Pi concentration of *osscl26* from leaf2 to leaf5, and the Pi concentration of *osscl26* in the older leaves and stems remained higher than the wild-type at 22 d (Figure 4E–G). At 28 d, the Pi concentration of *osscl26* in each leaf and stem showed a similar trend to the 22 d samples. However, the Pi concentration of *osscl26* in the older leaves decreased compared to the seedlings at 22 d under the LP and MP conditions (Figure 4H–J).

### 2.5. Expression of Phosphorus Homeostasis Genes of the osscl26 Mutant

To gain further insight into the role of OsSCL26 in regulating P homeostasis in rice, we investigated the expression levels of the phosphate starvation-induced genes in the shoot of the *osscl26* mutant under MP conditions. We selected several phosphate starvation-induced genes that were highly expressed above ground in rice. Our results showed that under MP conditions, the expression levels of *OsPHT1;1*, *OsPHT1;8*, *OsPHT2;1*, *OsPHT4;3*, *OsSPX-MFS1*, *OsSPX-MFS2*, and *OsNLA2* were upregulated in the *osscl26* mutant compared to the wild-type [10,12,25,26,27,28,29]. However, the expression of *OsSPX2* was downregulated in the *osscl26* mutant compared to the wild-type, and the expression levels of *OsSPX1* and *OsSPX4* did not differ significantly between the wild-type and the *osscl26* mutant (Figure 5).

### 2.6. Validation of AS Events in the Phosphorus-Related Genes of the osscl26 Mutant

To understand whether the mutations in *OsSCL26* affected the AS events of the P-related genes, we selected the P-related genes with multiple transcripts from the MSU database for validation using qRT-PCR. The results showed that the transcript abundance of *OsSPX-MFS2.1* was lower in the *osscl26* mutant than the wild-type, while the expression level of *OsSPX-MFS2.2* was higher than the wild-type (Figure 6A). Both the transcript abundances *OsNLA2.1* and *OsNLA2.2* were increased in the *osscl26* mutant (Figure 6B). Compared to the wild-type, the exclusion/inclusion ratios *OsSPX-MFS2* and *OsNLA2* were increased in the *osscl26* mutant (Figure 6C,D).

## 3. Discussion

In plants, the Serine/arginine-rich (SR) protein family is a group of eukaryotic splicing factors that play a critical role in the constitutive and selective splicing of precursor mRNAs [30,31]. The expression patterns and functions of the SR proteins vary between members [32,33]. Although the specific functions of OsSCL26 have not been systematically studied, we found through our experiments on its expression patterns that the *OsSCL26* gene was mainly expressed in old leaves and old leaf sheaths (Figure 2A), which was consistent with the growth phenotype of the *osscl26* mutant (Figure 3D,E). Under sufficient P conditions, the shoot height of the *osscl26* mutant was lower than that of the wild-type. Therefore, we considered that OsSCL26 plays an important role in rice growth and development.

The highly observed tissue expression of *OsSCL26* in the vascular bundles of stems and in the vascular sheaths of leaves (Figure 2B) suggests its role in long-distance P transport. This expression pattern is similar to that of *OsSCL25* [21]. Previous studies found that the mutation of OsSCL25 significantly affects the P uptake and accumulation in the shoot [21]. Therefore, we speculated that the function of the OsSCL subfamily is conserved in regulating the P distribution in the shoot.

The results of the phenotype, total P, and available P experiments of the *osscl26* mutant showed that the mutation of *OsSCL26* had no effect on the P content in the root of rice but increased the P content in the shoot. This was consistent with a previous T-DNA insertion line for *OsSCL26* [21]. Under low P conditions, the wild-type was dwarfed, but the *osscl26* mutant grew as well as under normal P conditions (Figure 3G,H). The accumulation of P in the shoot of the *osscl26* mutant alleviated the phenomenon of plant dwarfing (Figure 3D–F), indicating that the OsSCL26 protein regulated the growth of rice under deficient P conditions. Since the OsSCL26 protein was located in the stele of the roots and stems, we speculated that OsSCL26 was involved in the transport of P from underground to above ground. Through the experiment of the change in the P content in different tissues and different periods of rice, the results showed that the P content in the *osscl26* mutant was higher than that in the wild-type, which was mainly accumulated in the old leaves and did not transfer to the new leaves (Figure 4A–J). The redistribution of P from the old leaves to the new leaves through the stem of the *osscl26* mutant material was hindered, which affected the growth of the mutant material less than the wild-type, and limited growth and development. The experimental results were similar to the function of the OsSCL25 protein [21].

Numerous phosphate transporters and transcription factors that regulate the phosphate signaling pathways have been reported in rice. The phosphate transport family in rice mainly includes the PHT family and SPX domain-containing proteins [34,35]. Both OsPT1 and OsPT8 of the PHT family promote the transport of P from the roots to the shoots, and OsPT14 participates in the transport of P in chloroplasts [12,25,28]. OsSPX-MSF1 and OsSPX-MSF2 are involved in the transport of P in vacuoles, and the rice miR827 targets and regulates *OsSPX-MFS1* and *OsSPX-MFS2* [10,26]. In Arabidopsis, miR827 induced and mediated the transcription of *NLA1*, increasing the abundance of *PHT1* on the plasma membrane to accelerate the uptake of P under a P deficiency [36,37]. The expression of these P-related genes was significantly increased in the *osscl26* mutant (Figure 5A–J), suggesting that *OsSCL26* affects the absorption and transport of P at the transcriptional level. Therefore, we selected OsSPX-MFS2 and OsNLA2 for verification with multiple transcripts predicted by the MSU database. The results showed that the expression of *OsNLA2* increased in the different transcripts of the *osscl26* mutants. Although the expression of *OsSPX-MFS2.1* decreased in the *osscl26* mutants, the expression of *OsSPX-MFS2.2* significantly increased. *OsSPX-MFS2* and *OsNLA2* had significantly higher exclusion/inclusion ratios in the *osscl26* mutant, suggesting that *OsSCL26* was also involved in P homeostasis in rice at the post-transcriptional level, specifically regulating the mRNA splicing of the P-related gene. The detail underline mechanisms require further investigation.

The expression pattern, subcellular localization, and functional analysis of *osscl26* mutant materials provided valuable insight into the involvement of the OsSCL26 protein in how rice responds to a P deficiency. It serves as a crucial regulator of P homeostasis in rice, ensuring a balanced uptake and distribution of phosphorus within the plant.

## 4. Materials and Methods

### 4.1. Plant Materials and Growth Conditions

The wild-type Nipponbare (*Oryza sativa* L.) and single mutant *osscl26* created by CRISPR/Cas9 were used for all the experiments [38]. The plants were cultivated using hydroponics containing 0.18 mM (NH_4_)_2_SO_4_, 0.27 mM MgSO_4_, 0.09 mM KNO_3_, 0.18 mM CaNO_3_, 20.00 μM FeEDTA, 0.50 μM MnCl_2_, 3.00 μM H_3_BO_3_, 1.00 μM (NH_4_)_6_Mo_7_O_2_, 0.40 μM ZnSO_4_, and 0.20 μM CuSO_4_. The solution pH was adjusted to 5.5 and was renewed every two days. The seedlings were grown in a greenhouse in light: dark cycles for 14 h at 30 °C: 10 h at 25 °C, respectively. Three different Pi treatments, HP (high P; 450 µM), MP (sufficient P; 100 µM Pi), and LP (low P; 10 µM Pi) were applied in this study to operate the Pi uptake system shift from a low affinity to a high affinity [3,11].

### 4.2. Bioinformatics Analysis

The physicochemical characterization of the OsSCL26 protein was performed using the ExPASy-ProtParam tool (https://web.expasy.org/protparam/ (accessed on 28 April 2022)). The hydrophilicity/hydrophobicity profile of the protein was predicted using the ProtScale tool available on the ExPASy server (https://web.expasy.org/protscale/ (accessed on 28 April 2022)). The predictions for the transmembrane regions of the protein were carried out using the MHMM Server v. 2.0 tool (http://www.cbs.dtu.dk/services/TMHMM/ (accessed on 28 April 2022)). The conserved structural domains of OsSCL26 protein were predicted using the Structure-CDD tool (https://www.ncbi.nlm.nih.gov/Structure/cdd/wrpsb.cgi (accessed on 28 April 2022)). The NetPhos 3.1 Server software (http://www.cbs.dtu.dk/services/NetPhos/ (accessed on 28 April 2022)) was employed to predict the phosphorylation sites of the OsSCL26 protein. The SignalIP 3.0 Server software (http://www.cbs.dtu.dk/services/SignalP-3.0/ (accessed on 28 April 2022)) was utilized for the signal peptide prediction. The secondary structure of the OsSCL26 protein was predicted using the SOPMA program (https://npsa-prabi.ibcp.fr/cgi-bin/npsa_automat.pl?page=/NPSA/npsa_sopma.html (accessed on 28 April 2022)). The tertiary structure of the OsSCL26 protein was predicted using the SWISS-MODEL tool (https://swissmodel.expasy.org/ (accessed on 28 April 2022)). The amino acid sequences of the OsSCL26 proteins and other homologous proteins from the plants were downloaded from the NCBI database.

### 4.3. Gene Expression Analysis

To investigate the expression pattern of *OsSCL26* in the different growth stages for tillering, flowering, and filling, we exposed the seedlings of the wild-type rice to hydroponics 6 weeks and to a paddy field for 13 weeks. The roots, nodes, leaf blades, leaf sheaths, stems, panicles, and seeds were sampled for RNA extraction. The total RNA was extracted using an RNA extraction kit (TaKaRa) and cDNAs were synthesized using the RT-qPCR Master Mix (TaKaRa). A quantitative real-time PCR was performed using the following primer sets for *OsSCL26*. The relative expression of the genes of interest was normalized based on Actin using the 2^−ΔΔCT^ method.

For the alternative splicing (AS) analysis, the ratio was calculated using the following formula: exclusion/inclusion ratio = (1 − PSI)/PSI. The PCR primer sets were designed to ensure that either the spliced RNA or unspliced RNA was amplified. The first primer pair was designed to exactly cover the retained intron or alternative exon region for the alternative exon/intron inclusive transcripts, and the second primer pair amplified the splice junction region connecting the upstream flanking exon and the downstream flanking exon of the alternative exon/intron exclusive transcripts. The primers used for the qRT-PCR and AS analyses are listed in Appendix A.

### 4.4. Phenotypic Analysis of the osscl26 Mutant

The seedlings of the wild-type and *osscl26* mutant (10 days) were grown in nutrient solutions with three different P supplies, as mentioned above. After 24 days, the root and shoot lengths and dry weights were measured.

### 4.5. Histochemical Localization of the GUS Expression 

To determine the histochemical analysis of the *OsSCL26* expression, we amplified the 8,000 bp promoter region of *OsSCL26* into the vector p1300-GUS-Plus to generate *Pro_OsSCL26_*-1300-GUS. The fresh organs of *Pro_OsSCL26_*-1300-GUS were submerged in a GUS reaction mix (100 mM PBS (pH 7.0),10 mM Na_2_EDTA,1 mM K_3_[FeCN_6_],1 mM K_4_[FeCN_6_], 0.5% (*v*/*v*) Triton X-100, 20% (*v*/*v*) Methanol, 0.5 mg/mL X-Gluc, ddH_2_O) with vacuum infiltration for 30 min, and incubated at 37 °C overnight. After decoloring with 95% alcohol, the roots, shoots, and leaves were photographed using a stereo microscope for sectioning. The samples were then embedded in Agar and sectioned using a slicing machine. The slices were photographed using fluorescent microscopes. The primers used for the vector construction are listed in Appendix A.

### 4.6. Subcellular Localization of OsSCL26

To investigate the subcellular localization, we amplified the full-length open reading frame (ORF) of *OsSCL26* into the vector PHB-GFP and MCS-GFP to generate 35S::OsSCL26::PHB-GFP and 35S::OsSCL26::MCS-GFP, respectively. Once generated, 35S::OsSCL26::PHB-GFP was introduced into the strain EHA105α(*Agrobacterium tumefaciens*). The corresponding constructs were infections in tobacco leaves. Next, 35S::OsSCL26::MCS-GFP was transfected with DNA in a protoplast suspension. After the transfection, the cells were cultured in a protoplast medium overnight.

Then, the 35S::OsSCL26::PHB-GFP and 35S::OsSCL26::MCS-GFP fluorescence was detected using a confocal laser scanning microscope. The primers used for the quantitative real-time PCR are listed in Appendix A.

### 4.7. Nutrient Elements Concentration Analysis

The seedlings of the wild-type and *osscl26* mutant were grown in nutrient solutions with different P supplies for approx. 14 days, and their roots and shoots were harvested and dried at 105 °C for 15 min and 80 °C for 12 h, respectively. These samples were digested with HNO_3_/HClO_4_ (87:13 *v*/*v*) using a graphite disintegrator. After dissolving the samples in 2% HNO_3_, the P, Fe, Mn, Cu and Zn concentrations were determined using ICP-MS (PerkinElmer NexION 300X, Waltham, MA, USA).

### 4.8. Measurement of the Distribution of the Pi Concentration

The seedlings of the wild-type and *osscl26* mutant were grown in nutrient solutions with different P levels for approx. 14 days, and different leaves, roots and stems were harvested for the fresh samples. Inorganic P (Pi) was measured using the molybdate blue method [39]. 

### 4.9. Statistical Analyses 

The statistical analysis was performed using SPSS 19.0 for all the data. The significance differences were defined as (* *p* < 0.05, ** *p* < 0.01 and *** *p* < 0.001, Student’ s *t*-test).

## 5. Conclusions

In summary, the Serine/arginine-rich (SR) splicing factor OsSCL26 plays an important role in regulating P homeostasis in rice by influencing the P transport and distribution through the regulation of the transcription and splicing of the P transport genes. 

## Figures and Tables

**Figure 1 plants-12-02326-f001:**
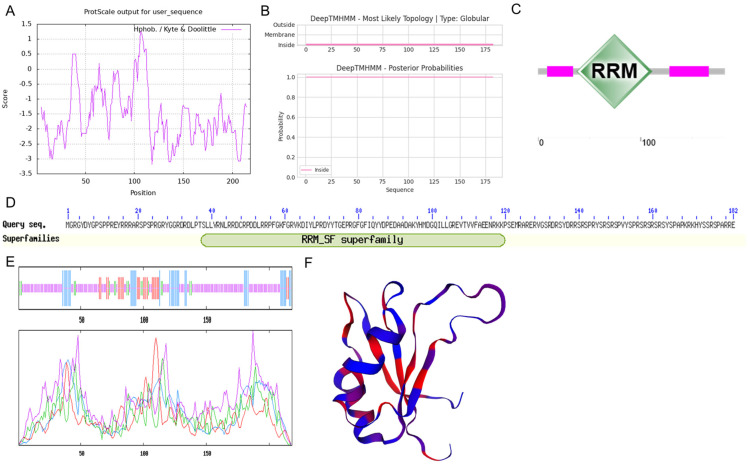
Protein properties of OsSCL26. The predictions of hydrophilicity/hydrophobicity in the (**A**), transmembrane (**B**), RRM domain (**C**,**D**), secondary structure (**E**), and tertiary structure (**F**) in OsSCL26.

**Figure 2 plants-12-02326-f002:**
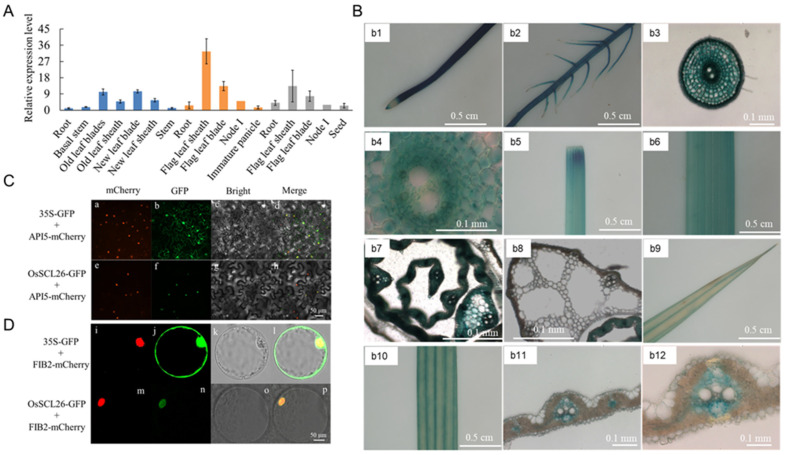
OsSCL26 expression pattern and its encoding protein subcellular localization. (**A**) Reverse transcription quantitative PCR (RT-qPCR) analysis of the relative expression levels of OsSCL26 at different tissues and growth stages. The data are given as the means ± the SD of three biological replicates. (**B**) GUS staining of various tissues in ProOsSCL26::GUS transgenic plants, including the root (**b1**,**b2**), cross section of the root mature zone (**b3**,**b4**), stem (**b5**,**b6**), stem cross section (**b7**,**b8**), leaf (**b9**,**b10**), and leaf cross section (**b11**,**b12**). Scale bars = 0.5 cm in (**b1**,**b2**,**b5**,**b6**,**b9**,**b10**) and 0.1 mm in (**b3**,**b4**,**b7**,**b8**,**b11**,**b12**). (**C**,**D**) The subcellular localization of OsSCL26. The empty vector and recombinant vector OsSCL26-GFP were co-expressed using OsAIP5-mCherry in the epidermal cells of tobacco leaves (**C**); GFP and the recombinant protein OsSCL26-GFP were co-expressed using FIB2-mCherry in rice protoplasts (**D**). The green signals indicate GFP, and the red signals indicate mCherry. Scale bars = 50 μm.

**Figure 3 plants-12-02326-f003:**
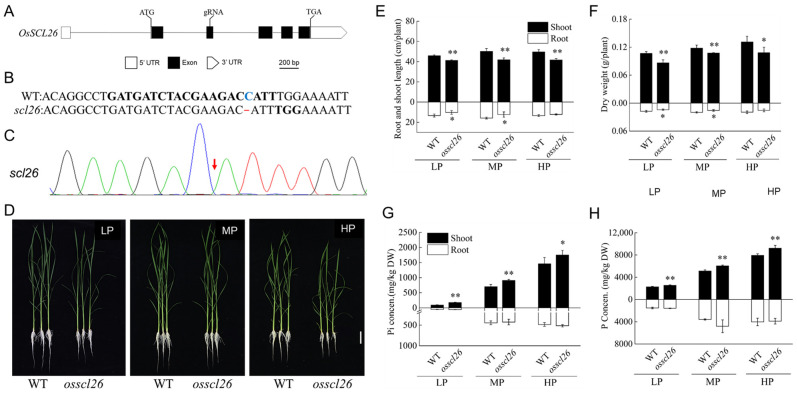
Mutation of *OsSCL26* changes in plant growth and Pi accumulation. (**A**) *OsSCL26* gene structure; the black lines and boxes show the introns and exons of *OsSCL26*, respectively, and the white box and wide arrow show the 5′UTR and 3′UTR of *OsSCL26*, respectively. The CRISPR/Cas9 target sites containing 21 bp are located in the second intron. (**B**,**C**) The results of the sequence from the *OsSCL26* mutation site. The *osscl26* mutant produced a 1 bp deletion of ‘C’ compared to the wild-type. (**D**) Images of the wild-type and mutant plants grown under LP (10 µM Pi), MP (100 µM Pi), and HP (450 µM Pi) conditions. Scale bars = 5 cm. (**E**) Root length and shoot height of the wild-type and mutant plants. (**F**) Root and shoot dry weight of the wild-type and mutant plants. (**G**,**H**) Pi concentrations (**G**) and total P concentrations (**H**) in the shoots and roots of the wild-type and mutant plants grown under LP, MP, and HP conditions. The values are the means ± the SD (*n* = 4). The values that were significantly different from the wild-type are indicated (* *p* < 0.05, and ** *p* < 0.01, Student’s *t*-test).

**Figure 4 plants-12-02326-f004:**
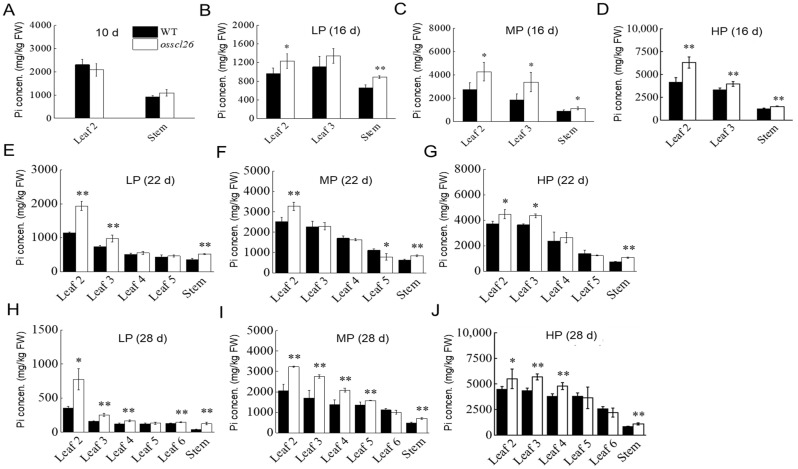
Distribution and accumulation of Pi in the wild-type plants and *osscl26* mutants in a 28 d time course. (**A**–**J**) Pi distribution in the indicated leaves and stem of the wild-type and *osscl26* mutant at 10 d (**A**), 16 d (**B**–**D**), 22 d (**E**–**G**), and 28 d (**H**–**J**). The values are the means ± the SD (*n* = 4). The rice seedlings were grown in a half Kimura B nutrient solution for 3 days, then transferred to the LP, MP, and HP conditions. Each leaf and stem of the wild-type and *osscl26* mutant were sampled at 10, 16, 22, and 28 d for the Pi concentration measurements. The values are the means ± the SD (*n* = 4). The values that were significantly different from the wild-type are indicated (* *p* < 0.05, ** *p* < 0.01, Student’s *t*-test).

**Figure 5 plants-12-02326-f005:**
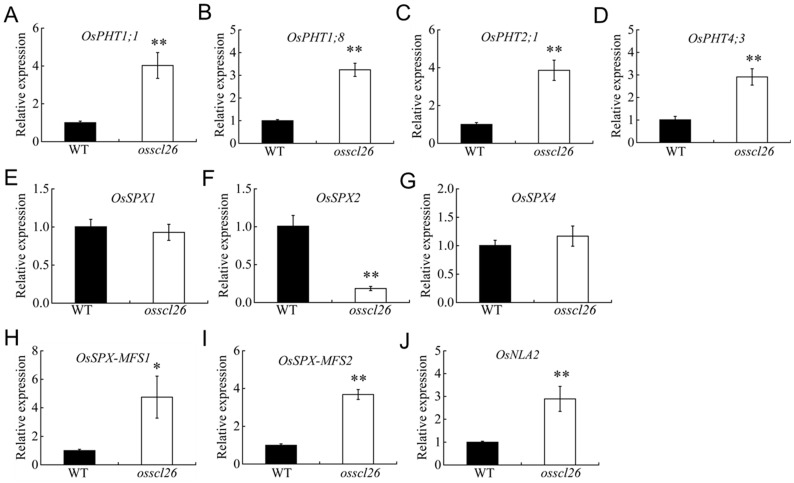
Expression of the phosphorus homeostasis genes in the wild-type plants and *osscl26* mutants. The fourteen-day-old seedlings grown under +P condition shoots were sampled and used for RNA extraction. The expression of *OsPHT1;1* (**A**), *OsPHT1;8* (**B**), *OsPHT2;1* (**C**), *OsPHT4;3* (**D**) *OsSPX1* I, *OsSPX2* (**F**), *OsSPX4* (**G**), *OsSPX-MFS1* (**H**), *OsSPX-MFS2* (**I**), and *OsNLA2* (**J**) were determined using quantitative real-time PCR. *OsActin1* was used as the internal standard. The values are the means ± the SD (*n* = 4). The values that were significantly different from the wild-type are indicated (* *p* < 0.05, and ** *p* < 0.01, Student’s *t*-test).

**Figure 6 plants-12-02326-f006:**
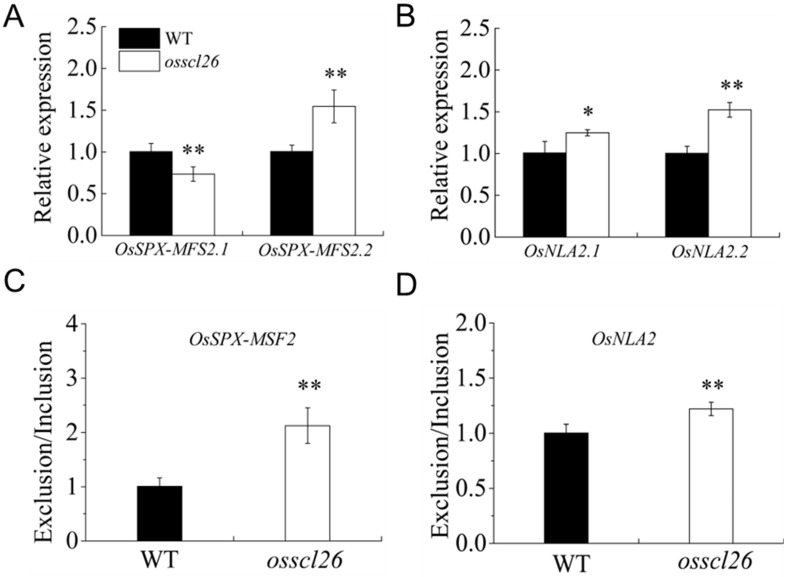
Validation of AS events of phosphorus-related genes in the *osscl26* mutant. The fourteen-day-old seedlings grown under +P condition shoots were sampled and used for RNA extraction. (**A**) Relative expression of *OsSPX-MFS2.1* and *OsSPX-MFS2.2* (the first and the second predicted transcripts of *OsSPX-MFS*2 in the MSU database). (**B**) Relative expression of *OsNLA2.1* and *OsNLA2.2* (the first and the second predicted transcripts of *OsNLA2* in the MSU database). (**C**,**D**) The exclusion/inclusion ratio of *OsSPX-MFS*2 (**C**) and *OsNLA2* (**D**). The values are the means ± the SD (*n* = 3). The values that were significantly different from the wild-type are indicated (* *p* < 0.05, and ** *p* < 0.01, Student’s *t*-test).

## Data Availability

All data are presented in the manuscript.

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
