# Peer review of "The Splicing Factor OsSCL26 Regulates Phosphorus Homeostasis in Rice"

_plants, 2023, doi:10.3390/plants12122326_

Round 1

Reviewer 1 Report

Comments for MS No. plants-2416291:

In soils, the concentration of soluble P seldom exceeds 10 μM and is much lower in the rhizosphere. Toward to poor Pi availability in soils, plants have evolved well-regulated systems for Pi acquisition and recycling, including the high- and low-affinity Pi transporters and the ability to induce root architectural changes to P. The research area of the manuscript, plants-2416291 entitled ‘Splicing Factor OsSCL26 Regulates Phosphorus Homeostasis in Rice’ is very good. However, the manuscript has many concerns and may please be addressed by the authors before acceptance.

Major concern:

1.Less than10 μM soluble P in the pot should be taken for the –P condition study.

2. Are you sure that the experimental material does not carry the Pup1/PISTOL allele in the background. This allele has also effects under low phosphorus condition. It is better if you are taking material that should not have this gene in the background.

3. At line 171, under -P and +P conditions, Pi concentration of osscl26 mutant in the leaf2 and stem were higher than wild type in 16 d. How you are sure of p alleviation in osspl26 mutants as under both situations Pi is more!!

4. At line 172, under ++P condition, Pi concentration of osscl26 mutant in the leaf2, leaf3 and stem were higher than wild type in 16 d. How it is alleviating osspl26 which seems to be efficient under higher P also.

5. At line 177, Pi concentration of osscl26 mutant in older leaves were decreased than seedlings in 22 d under -P and +P condition. How the osspl26 mutant is transporting to the younger leaves?

6. At line 270, the expression of OsSPX-MFS2.1 was decreased in the osscl26 mutants. How do you justify this result in the osspl26 mutants?

Minor concern:

1.At line259 to 261: Both OsPT1 259 and OsPT8 of the PHT family promote the transport of P from roots to shoots, and OsPT14 participates in the transport of P in chloroplasts. SPX-MSF1 and SPX-MSF2 are involved in the transport of P in vacuoles. Please cite reference for these statements?

2.At line No. 281, Single mutant of osscl26 created by CRISPR/Cas9 was used for all experiments. Please provide the reference.

3.At line No. 282, plants were grown instead of plants were growth.

4.At line no.304, ‘we ‘ is repeated.

5.At line No. 324, the primers used for qRT-PCR are listed in Table S1.

6.At line No. 339, harvested and dried instead of harvested to dried

7.Check whole manuscript for gap before the bracket [].

8.Check whole manuscript for use of comma.

9.Check whole manuscript for use of gap after full stop.

Good

Author Response

We appreciate very much the reviewer’s constructive comments and valuable suggestions which would help us to improve the quality of our manuscript. revised the paper according to the comments as described below.

Reviewer: 1

Major concern:

1.Less than10 μM soluble P in the pot should be taken for the –P condition study.

Response: In phosphorus nutrient study in rice, 10 μM soluble P was widely used for low P treatment, see references (Plant Physiol. 2012;159:169-83. doi: 10.1104/pp.112.194217; Plant Cell, 2020; 32:740-757. doi: 10.1105/tpc.19.00685). To make it clear, we used the abbreviation LP (low P) instead of -P to indicate the 10 μM Pi treatment. Accordingly, MP (middle P: 91 μM Pi), LP (low P: 10 μM Pi) and HP (high P: 450 μM Pi) in the revised manuscript.

  1. Are you sure that the experimental material does not carry the Pup1/PISTOL allele in the background. This allele has also effects under low phosphorus condition. It is better if you are taking material that should not have this gene in the background.

Response: Yes, we used the japonica cultivar Nippon bare does not carry the Pup1/PISTOL allele. In the other hand, in this study we focus on the splicing factor mediated regulation in P homeostasis, this is independent of Pup1/PISTOL allele effects.

  1. At line 171, under -P and +P conditions, Pi concentration of osscl26 mutant in the leaf2 and stem were higher than wild type in 16 d. How you are sure of p alleviation in osscl26 mutants as under both situations Pi is more!!

Response: The results showed that under LP and MP conditions, the osscl26 mutant accumulated higher P in the leaf2 and stem than the wild type control,  suggested that OsSCL26 was involved in negative regulation of P distribution  and re-distribution in rice, this is consistent with our previous observation that SCL subfamily of SR protein may participate in P transport regulation (Dong et al., 2018). Also, our gene expression analysis further demonstrate this point.

  1. At line 172, under ++P condition, Pi concentration of osscl26 mutant in the leaf2, leaf3 and stem were higher than wild type in 16 d. How it is alleviating osspl26 which seems to be efficient under higher P also.

Response: This is similar with previous comments, in addition to the role of OsSCL26 in P transport regulation, the mutation of OsSCL26 lead to high accumulation of P in leaves, thus under HP conditions, heavier hyper-sensitivity phenotype was observed for the mutant.

  1. At line 177, Pi concentration of osscl26 mutant in older leaves were decreased than seedlings in 22 d under -P and +P condition. How the osspl26 mutant is transporting to the younger leaves?

Response: The results suggested that in the osscl26 mutant, P could be distributed/redistributed to younger leaves through PT, PHO1 genes although their contribution may less as compared with WT control.

  1. At line 270, the expression of OsSPX-MFS2.1 was decreased in the osscl26 mutants. How do you justify this result in the osspl26 mutants?.

Response: The results showed that the expression of OsSPX-MFS2.1 was decreased in osscl26 mutants, while the expression of OsSPX-MFS2.2 was significantly increased suggested that in the OsSCL26 could regulated P homeostasis through regulation the distribution of P in cytosol and vacuole, and also directly affect the P signaling pathway. This is very interesting although the detail mechanisms underline awaits further analyze.

Minor concern:

  1. At line259 to 261: Both OsPT1 259 and OsPT8 of the PHT family promote the transport of P from roots to shoots, and OsPT14 participates in the transport of P in chloroplasts. SPX-MSF1 and SPX-MSF2 are involved in the transport of P in vacuoles. Please cite reference for these statements?

Response: We have added citing references, see line 263 in the revised manuscript.

  1. At line No. 281, Single mutant of osscl26 created by CRISPR/Cas9 was used for all experiments. Please provide the reference.

Response: We have added citing references, see line 284 in the revised manuscript.

  1. At line No. 282, plants were grown instead of plants were growth.

Response: We have revised the sentence as suggested.

  1. At line no.304, ‘we ‘ is repeated.

Response: We have deleted the repeated” we”.

  1. At line No. 324, the primers used for qRT-PCR are listed in Table S1.

Response: We have revised the sentence as suggested.

  1. At line No. 339, harvested and dried instead of harvested to dried

Response: We have revised the sentence as suggested.

  1. Check whole manuscript for gap before the bracket [].

Response: We have checked the whole manuscript, and revised accordingly.

  1. Check whole manuscript for use of comma.

Response: We have checked the using of comma in whole manuscript, and revised accordingly.

9.Check whole manuscript for use of gap after full stop.

Response: We have checked the using of gap in whole manuscript, and revised accordingly.

Reviewer 2 Report

In general, it is interesting to study the molecular mechanisms of P homeostasis in rice. But some important points have to be clarified. Herewith I would like to suggest a major revision in the manuscript. I believe the paper could be further strengthened by added information about the following points:

Comment 1: Lines 90-93: Numbers one to nine without units should be presented as words; numbers 10 and over without units should be presented as numerals.

Comment 2: The data presented Fig. 3E-F and Fig. 4 are in low quality.

Comment 3: Why there is only one genotype in KO transgenic events, It would be better to have two transgenic events of different genotype.

Comment 4: Lines 183-184: “(A-J) Pi distribution in the indicated leaves, root, and stem of the wild type and osscl26 mutant at 10 d (A), 16 d (B-D), 22 d (E-G), and 28 d (H-J)”. I did not find the Pi distribution of root in Figure 4.

Comment 5: The authors show that the dry weight of shoots and roots in osscl26 mutants were significantly lower compared to the wild type. We then measured the inorganic P (Pi) and total P concentrations of the osscl26 mutant and results showed that both the Pi and total P is higher in the shoots was significantly higher in the mutant than that of the wild type. The authors should perform the analysis of Pi content in root and shoots. I wonder whether the increase in P concentration is simply due to the enrichment effect caused by the decrease in biomass.

Comment 6: Lines 203-204: “The expression of OsPHT;1 (A), OsPHT;8 (B), OsSPX-MFS2 (J), and OsNLA-L (J)”. Genes should be consistent with those shown in Figure 5.

Comment 7: I would like to recommend Authors 'rewrite the conclusion'. The conclusion was extremely similar to the abstract.

Comment 8: References of 33-34 are not cited in the article.

Comment 9: Table S1 should be shown in the normal format.

Comment 10: All genes in manuscript should be italic, please check and correct it.

Comment 11: The entire manuscript should be professionally edited for language usage and grammar by English-native experts. It is better to secure help from the professional English-native experts.

The entire manuscript should be professionally edited for language usage and grammar by English-native experts. It is better to secure help from the professional English-native experts.

Author Response

Reviewer: 2

Comment 1: Lines 90-93: Numbers one to nine without units should be presented as words; numbers 10 and over without units should be presented as numerals.

Response: We have revised the numbers in the sentence as suggested.

Comment 2: The data presented Fig. 3E-F and Fig. 4 are in low quality.   

Response: We have re-prepare the Fig. 3E-H and Fig. 4. as suggested. the quality of the figures had improved significantly. See the new figures in the revised manuscript.

Comment 3: Why there is only one genotype in KO transgenic events, It would be better to have two transgenic events of different genotype.

Response: We have reported another mutant line (T-DNA insertion) of OsSCL26 in our previous paper (Dong et al., 2018), they showed similar high P accumulation phenotype, thus in this paper, we did detail analysis of P distribution in the new CRISPR line. We have added the discussion in the discussion part, see line 249 in the revised manuscript.

Comment 4: Lines 183-184: “(A-J) Pi distribution in the indicated leaves, root, and stem of the wild type and osscl26 mutant at 10 d (A), 16 d (B-D), 22 d (E-G), and 28 d (H-J)”. I did not find the Pi distribution of root in Figure 4..

Response: We have find that P accumulation mainly observed in the shoot part, see result in figure 3, thus we didn’t show root P concentration in fig 4, we delete root in the figure legend in the revised manuscript.

Comment 5: The authors show that the dry weight of shoots and roots in osscl26 mutants were significantly lower compared to the wild type. We then measured the inorganic P (Pi) and total P concentrations of the osscl26 mutant and results showed that both the Pi and total P is higher in the shoots was significantly higher in the mutant than that of the wild type. The authors should perform the analysis of Pi content in root and shoots. I wonder whether the increase in P concentration is simply due to the enrichment effect caused by the decrease in biomass.

Response: Actually, we did measured the root and shoot P concentrations in mutant and wild type, results showed that difference in P accumulations were only observed in the shoot part, thus we focus on the shoot P distribution in the following analyze, see Figure 3 and 4.

Comment 6: Lines 203-204: “The expression of OsPHT;1 (A), OsPHT;8 (B), OsSPX-MFS2 (J), and OsNLA-L (J)”. Genes should be consistent with those shown in Figure 5.

Response: We have revised the numbers in the sentence as suggested. See line 194-195 in the revised manuscript.

Comment 7: I would like to recommend Authors 'rewrite the conclusion'. The conclusion was extremely similar to the abstract.

Response: As suggested, we re-write the conclusion in the revised manuscript,  see line 354-356 in the revised manuscript.

Comment 8: References of 33-34 are not cited in the article.

Response: Reference 33 was cited (line 231), 34 was deleted from the manuscript.

Comment 9: Table S1 should be shown in the normal format.

Response: We have re-formatted the Table S1 as suggested.

Comment 10: All genes in manuscript should be italic, please check and correct it.

Response: We have checked all genes used in the manuscript and revised to italic.

Comment 11: The entire manuscript should be professionally edited for language usage and grammar by English-native experts. It is better to secure help from the professional English-native experts.

Response: We have used the professionally proofread service to edit the manuscript.

Round 2

Reviewer 2 Report

The author gives a serious answer to the question raised last time. At this stage, I think the article is acceptable. But English writing needs further improvement.

The author gives a serious answer to the question raised last time. At this stage, I think the article is acceptable. But English writing needs further improvement

Author Response

Thank you very much for your comments, as suggested, I did further English editing to this manuscript.